# Learning from Teaching Regularization: Generalizable Correlations Should be Easy to Imitate

**Can Jin**[1*]  **Tong Che**[2*]  **Hongwu Peng**[3†]

**Yiyuan Li**[4†]  **Dimitris N. Metaxas**[1‡]  **Marco Pavone**[5‡]

[1]Rutgers University  [2]Nvidia Research  [3]University of Connecticut
[4]University of North Carolina at Chapel Hill  [5]Stanford University
can.jin@rutgers.edu, tongc@nvidia.com

## Abstract

Generalization remains a central challenge in machine learning. In this work, we propose *Learning from Teaching* (**LoT** ), a novel regularization technique for deep neural networks to enhance generalization. Inspired by the human ability to capture concise and abstract patterns, we hypothesize that generalizable correlations are expected to be easier to imitate. LoT operationalizes this concept to improve the generalization of the main model with auxiliary student learners. The student learners are trained by the main model and, in turn, provide feedback to help the main model capture more generalizable and imitable correlations. Our experimental results across several domains, including Computer Vision, Natural Language Processing, and methodologies like Reinforcement Learning, demonstrate that the introduction of LoT brings significant benefits compared to training models on the original dataset. The results suggest the effectiveness and efficiency of LoT in identifying generalizable information at the right scales while discarding spurious data correlations, thus making LoT a valuable addition to current machine learning. Code is available at https://github.com/jincan333/LoT.

## 1 Introduction

Improving the generalization performance of models on unseen data is a major challenge in machine learning [6, 7, 73, 79, 107]. Despite its significant advances, identifying the most generalizable model within the vast space of potential models remains challenging. Existing deep learning approaches focus on crafting the hypothesis spaces where prediction errors are optimized using training data [33, 69, 71]. These spaces are shaped by inductive biases [33, 70] embedded in the neural architectures which include implicit assumptions about the data [1, 25, 95], objective functions (notably regularizers) [20, 68, 103], and learning methodologies [14, 72, 87].

In this paper, to enhance generalization, we use the methodology of regularization [37, 51, 88], which prioritizes specific regions in the hypothesis spaces. Regularization techniques often involve employing auxiliary losses or regularizers [20, 38, 103] alongside the primary task losses. For instance, L1 regularization [41, 92, 93] encourages sparsity within models [16, 40, 54, 57]. Other regularization techniques include model averaging [44, 102], dropout techniques [37, 67, 100], and additional

---

[0*]Equal contribution, [†]Equal contribution, [‡]Equal advising, Correspondence to: Can Jin <can.jin@rutgers.edu>, Tong Che <tongc@nvidia.com>.

optimization components [43, 63, 104]. Due to its effectiveness and simplicity, regularization is critical in modern machine learning techniques for achieving better generalization [37, 109].

We aim to answer the research question: *Among all possible models fitting the training data, which ones are inherently generalizable?* A common belief in cognitive science is that human intelligence development involves distilling information and filtering out extraneous details to discern 'simple' correlations among a few selected relevant abstract variables [18, 94]. This approach leads to the formation of correlations through simple patterns [2, 56] at the right scales. However, identifying simple correlations in deep learning remains challenging, mostly due to not being easy to identify the right scale of the problem. Studies in emergent languages suggest that the more structured a language is, the more efficiently it can be transmitted to message receivers [11, 56]. Inspired by this finding, we propose defining simple and generalizable correlations at the right scales, as those that can be readily imitated by other learners, provided they possess suitable inductive biases.

Based on this definition, we propose a novel regularization approach, *Learning from Teaching* (**LoT**). The core of LoT is to compute a measure of 'imitability' for the main model to learn data correlations at the correct scales. By adding this measure to the objective function and optimizing it during training, we encourage the teacher model to refine its learned multiscale correlations, making them more accessible through teaching, which in turn leads to better generalization. LoT computes this measure by jointly training the main model as the 'teacher' with one or more auxiliary 'student' models. The student models strive to distill and assimilate the correlations acquired by the teacher model. Thus, the learning performance of the student defines the measure of imitability of the teacher, which is then used as the LoT regularizer.

We conduct comprehensive experiments using LoT to improve the Reinforcement Learning (RL) formulation, as well as in Natural Language Processing (NLP) and Computer Vision (CV) applications. In RL, the experimental results demonstrate that LoT attains an average normalized reward enhancement of $44\%$ on four Atari games. In language modeling tasks, LoT achieves significant perplexity reductions on the Penn Tree Bank [64] and WikiText-103 [65]. Notably, LoT enhances the supervised fine-tuning performance of LLaMA [96, 97] models on GSM8K [19] and MATH [35]. In image classification tasks, LoT achieves accuracy gains of $1.99\%$ and $0.83\%$ on CIFAR-100 [49] and ImageNet-1K [23], respectively.

## 2 Methodology

### 2.1 Generalizable and Spurious Correlations

Given a dataset $\mathcal{D} = \{(\mathbf{x}_1, y_1), \cdots, (\mathbf{x}_n, y_n)\}$ generated from a data-generating distribution $\hat{D}$, there are infinitely many continuous functions $f$ such that $f(\mathbf{x}) = y$ for all $(\mathbf{x}, y) \in \mathcal{D}$. Therefore, finding the $f$ that precisely models the true generalizable correlation between $\mathbf{x}$ and $y$ is challenging, especially with real-world data like natural images, which are complex and multiscale. In such scenarios, a neural network may compute incorrect (according to the ground-truth relationship between variables) yet perfect (in the empirical data distribution) correlations that explain the relationship between $\mathbf{x}$ and $y$ [32, 73]. This phenomenon is particularly evident when $y$ is entirely noise-based and independent of $\mathbf{x}$, but the neural network still fits $y$ to $\mathbf{x}$ perfectly [73, 107]. This process, often called brute-force memorization [2, 13], involves the network creating intricate computational strategies to encode all $(\mathbf{x}, y)$ pairs in the samples. Consequently, correlations established in this way are spurious, originating from sampling noise in the data rather than ground-truth relationships.

But how do humans distinguish generalizable correlations from spurious ones? Instead of relying on brute-force memorization to establish input-output correspondences, humans naturally focus on understanding high-level concepts within the input data, selectively ignoring irrelevant details [18, 94]. This approach leads to the formation of correlations through simple, comprehensible patterns [2, 11]. Empirical evidence in emergent languages also suggests that the more compositional a language is, the more learners will use it [11, 56].

We can, therefore, define the distinctions between generalizable and spurious correlations. First, generalizable correlations are simple and comprehensible, exhibiting lower Kolmogorov Complexity [31, 58, 90]. Second, while there is only one ground-truth correlation for a dataset, the number of spurious correlations can be massive. These two major distinctions lead to the following hypothesis.

**Hypothesis:** Generalizable correlations should be more easily imitable by learners compared to spurious correlations. Specifically, assume $T_G$ and $T_S$ are two teacher models that capture the generalizable correlation and spurious correlation from a dataset, respectively. We have student learners $S_G$ and $S_S$ that separately imitate $T_G$ and $T_S$:

- From an effectiveness perspective, the final training and test losses of learner $S_G$ after training are typically lower than those of learner $S_S$.
- From an efficiency perspective, during training, the test losses of learner $S_G$ decrease more rapidly than those of $S_S$.

This hypothesis emphasizes that generalizable correlations inherent in data are not only more interpretable but also more readily imitable. It suggests that the inherent simplicity and uniqueness of generalizable correlations make them more attainable and recognizable for learning algorithms, in contrast to the complex and abundant nature of spurious correlations derived from noise. In the following we present our novel approach.

## 2.2 Learning from Teaching Regularization

Building upon the Hypothesis, we propose that the ease of imitation of the teacher model by student models can serve as a proxy for the generalizability of learned representations. By measuring the 'imitability' of the teacher model in the learning process, we can infer the generalizability of it. A teacher that is easier to imitate implies higher generalization. We then design a novel regularization approach that involves training a teacher model $T$ alongside student models $S$ to imitate $T$, subsequently measuring the imitability of the teacher during training. We maximize imitability by incorporating it as an additional loss during the training of the teacher $T$. This imitability loss is termed the Learning from Teaching regularizer (LoT regularizer). By doing so, $T$ is optimized to be a teacher that is easier to imitate and, thus, possesses superior generalization compared to models without the LoT regularizer. We refer to this class of regularization methods as 'Learning from Teaching Regularization' (LoT). LoT aligns with the broader concept of regularization in machine learning, where the goal is to promote generalizable representations and prevent overfitting.

Although LoT can be applied to supervised, unsupervised, and reinforcement learning, we begin our discussion with supervised learning. We train a network $T_{\boldsymbol{\theta}}$, parameterized by $\boldsymbol{\theta}$, as the main model, which also serves as the teacher model. Additionally, we train a set of $K$ networks $S_i, i = 1, 2, \cdots, K$, as the student models[1]. The total set of parameters of the $K$ networks is denoted by $\boldsymbol{\phi}$. Given a training dataset $\mathcal{D}_t = \{(\mathbf{x}_1, y_1), \cdots, (\mathbf{x}_n, y_n)\}$, we train $T$ and $S$ to model $p(y|\mathbf{x})$, denoted as $p_t(y|\mathbf{x})$ and $p_s(y|\mathbf{x})$, respectively. Additionally, LoT includes a predefined imitability metric $\mu_{s,t}(\cdot) = \mu(S(\cdot), T(\cdot))$. Intuitively, $\mu_{s,t}$ measures the difference between $S$ and $T$'s predictions on the same input (occasionally denoted as $\mu$ henceforth for convenience). There are many possible choices for the metric $\mu$, such as the $L^2$ loss between the hidden representations of a specific layer. In our experiments, we choose $\mu(\mathbf{x}) = \mu_{\mathrm{KL}}(p_s(y|\mathbf{x})||p_t(y|\mathbf{x}))$, which is the KL-divergence [21], to quantify the distribution similarity between $S$ and $T$.

We first train the teacher model. The objective function of the teacher combines the regular task loss with the additional LoT regularizer $R(\boldsymbol{\theta})$ (defined in Equation 3). For example, in supervised learning, we can use the negative log-likelihood loss for the regular task loss, and the objective function can be written as:

$$L_t(\boldsymbol{\theta}) = -\frac{1}{|\mathcal{D}_t|} \sum_{(\mathbf{x}_i, y_i) \in \mathcal{D}_t} \log p_t(y_i|\mathbf{x}_i) + R(\boldsymbol{\theta}), \qquad (1)$$

where $|\mathcal{D}_t|$ is the number of samples in the dataset $\mathcal{D}_t$.

To train the student networks and enhance information diversity, we require an independent unlabelled dataset, denoted as $\mathcal{D}_s = \{\mathbf{x}_1, \cdots, \mathbf{x}_m\}$. This dataset can be identical to $\mathcal{D}_t$, or generated either by a generative model trained on $\mathcal{D}_t$ or through alternative augmentation methods (e.g. synthetic data generation). This unlabelled dataset constitutes the environment for the student networks to follow the prediction of the teacher and, therefore, explores and generalizes beyond the original training data.

---

[1]For convenience, $S$ is referred to as a single student learner henceforth.

Specifically, the student networks' goal is to imitate the correlations acquired by the teacher network during the training process. The training loss for students can be written as:

$$L_s(\boldsymbol{\phi}) = \frac{1}{|\mathcal{D}_s|} \sum_{\mathbf{x} \in \mathcal{D}_s} \sum_{i=1}^{K} \mu_{s_i,t}(\mathbf{x}), \tag{2}$$

where $|\mathcal{D}_s|$ is the number of samples in the unlabelled dataset $\mathcal{D}_s$. The loss function $L_s$ encourages the student networks to learn from the teacher network by minimizing the difference between their predictions, as measured by the metric $\mu_{s,t}(\mathbf{x})$.

The feedback from all students $S_i$ constitutes the LoT regularizer:

$$R(\boldsymbol{\theta}) = \frac{\alpha}{|\mathcal{D}_s|} \sum_{\mathbf{x} \in \mathcal{D}_s} \sum_{i=1}^{K} \lambda_i \mu_{t,s_i}(\mathbf{x}), \tag{3}$$

where $\lambda_i \geq 0$ represents the coefficient weight of the $i$-th student $S_i$, with $\sum_{i=1}^{K} \lambda_i = 1$. The $\lambda_i$ can be either a learnable parameter or fixed, such as $\frac{1}{K}$. Essentially, the LoT regularizer measures the imitability of the teacher. The regularization coefficient $\alpha$ controls the trade-off between the original task learning objective of $T$ and the feedback from the students.

The detailed procedure of LoT for supervised and unsupervised learning is outlined in Algorithm 1, and LoT regularization for RL (using PPO as an example) is outlined in Algorithm 2. The teacher $T$ and student $S_i$ networks are initialized differently to ensure they learn diverse features and representations. In both algorithms, the teacher and student networks iteratively learn from each other, with the students imitating the teacher's correlations and the teacher incorporating the students' feedback into the learning process.

---

**Algorithm 1** Learning from Teaching Regularization

1: **Input:** Dataset $\mathcal{D}_s, \mathcal{D}_t$, Regularization Coefficient $\alpha > 0$, Student Steps Ratio $N > 0$
2: Initialize teacher network $T$ parameterized by $\boldsymbol{\theta}$ and student networks $S_i, i = 1, 2, \cdots K$, parameterized by $\boldsymbol{\phi}$.
3: **repeat**
4:     Sample a batch of data $\mathcal{B}_t \subset \mathcal{D}_t, \mathcal{B}_s \subset \mathcal{D}_s$
5:     Compute $\tilde{R}(\boldsymbol{\theta}) = \frac{\alpha}{|\mathcal{B}_s|} \sum_{\mathbf{x} \in \mathcal{B}_s} \sum_{i=1}^{K} \lambda_i \mu_{t,s_i}(\mathbf{x})$
6:     Compute $\tilde{L}_t(\boldsymbol{\theta}) = -\frac{1}{|\mathcal{B}_t|} \sum_{(\mathbf{x},y) \in \mathcal{B}_t} \log p_t(y|\mathbf{x}) + \tilde{R}(\boldsymbol{\theta})$
7:     Update $\boldsymbol{\theta}$ using gradient $\nabla_{\boldsymbol{\theta}} \tilde{L}_t(\boldsymbol{\theta})$
8:     **for** $i = 1$ **to** $N$ **do**
9:         Sample $\mathcal{B}_s \subset \mathcal{D}_s$
10:        Compute $\tilde{L}_s(\boldsymbol{\phi}) = \frac{1}{|\mathcal{B}_s|} \sum_{\mathbf{x} \in \mathcal{B}_s} \sum_{i=1}^{K} \mu_{s_i,t}(\mathbf{x})$
11:        Update student networks' parameters $\boldsymbol{\phi}$ using loss gradient $\nabla_{\boldsymbol{\phi}} \tilde{L}_s(\boldsymbol{\phi})$
12:     **end for**
13: **until** $T$ converges

---

## 2.3 Discussion

The works most related to LoT are knowledge distillation (KD) [29, 36] and ease-of-teaching [11, 56] in emergent languages. However, LoT differs significantly from these approaches. In KD, a teacher model containing task-specific knowledge transmits this knowledge to a student model (often smaller than the teacher), with the primary focus on the student's performance post-distillation. Conversely, in LoT , both the teacher and student models may lack or possess different task-specific knowledge. Generalization is improved through joint training, incorporating additional signals from student feedback. In emergent languages, Li and Bowling [56] propose that structured language is easier to teach to other agents than less structured ones, achieving higher task success rates with less training. Additionally, Chaabouni et al. [11] identify a strong positive correlation between language transmission efficiency to new message receivers and the degree of compositionality (structuredness) of the language. In LoT , we focus on tasks distinct from emergent languages, finding that generalizable correlations are easier to imitate. Under our Hypothesis, we design a novel

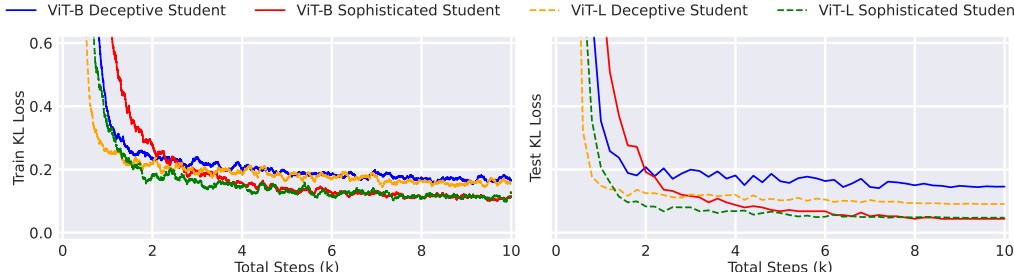

Figure 1: Training and test KL-divergence losses of student models in LoT using ViT-B/16 and ViT-L/16 on CIFAR-100 with different teacher models. The sophisticated students achieve lower losses than the deceptive students given the same computational budget.

LoT regularizer and algorithm to enhance the generalization of deep neural networks, extending the ease-of-teaching concept to supervised, unsupervised, and reinforcement learning. In parallel work, Ning et al. [74] proposes Learning by Teaching (LbT), which utilizes teacher and student models to generate answers as training samples for the teacher model. However, the regularization method in LoT is fundamentally distinct from that in Ning et al. [74].

## 3  Experiments

We first validate our Hypothesis in Section 3.1. Subsequently, we assess the performance of LoT across several tasks: Atari games (Section 3.2), language modeling (Section 3.3), and image classification (Section 3.4). We compare LoT to a Teacher-only baseline, wherein the regularization coefficient $\alpha$ in $R(\boldsymbol{\theta})$ is set to 0, thereby blocking the student feedback. Unless specified otherwise, we employ only one student model. Except for the Atari games where the student can learn from the offline samples of the teacher, we set $N = 1$ to manage computation (we study the impact of $N$ in Section 3.6). Moreover, we study the computational efficiency and effects of hyperparameters of LoT in Sections 3.5 and 3.6.

### 3.1  Generalizable Correlations are Easier to Imitate than Spurious Correlations.

In our Hypothesis, learners are presumed to more readily imitate generalizable correlations than spurious ones. To investigate this, we design experiments involving two distinct teacher models: a sophisticated teacher and a deceptive teacher. The sophisticated teacher effectively captures generalizable correlations, while the deceptive teacher primarily learns spurious correlations. We use an identical student model to learn from both teachers separately, monitoring the student-teacher KL divergence during training and testing. The student that learns easier-to-imitate correlations is expected to exhibit lower training and test KL losses with fewer training steps.

We employ the ViT-B/16 and ViT-L/16 architectures [24] for both the teachers and students. The sophisticated teachers are trained on the full CIFAR-100 [49] training set for $10,000$ steps to achieve optimal convergence. The deceptive teachers, using the same hyperparameters and training steps as the sophisticated teachers, are trained on a random subset of $2,560$ images from the CIFAR-100 training set, leading to over-fitting. Consequently, the sophisticated teachers are expected to exhibit better generalization ability (their test accuracy surpasses that of the deceptive teachers by $14\%$).

The two student models referred to as the sophisticated student and the deceptive student, share identical hyperparameters and initializations. They are trained to imitate the correlations from their respective teachers on the full CIFAR-100 training set. The teacher models are kept frozen during the training of the students, with the objective $L_s(\boldsymbol{\phi})$ defined as follows:

$$L_s(\boldsymbol{\phi}) = \frac{1}{|\mathcal{D}_s|} \sum_{\mathbf{x} \in \mathcal{D}_s} \mu_{\text{KL}}(p_s(y|\mathbf{x})||p_t(y|\mathbf{x})), \tag{4}$$

where $\mathcal{D}_s$ represents the full training set of CIFAR-100.

We present the training and test losses in Figure 1 and make the following observations:

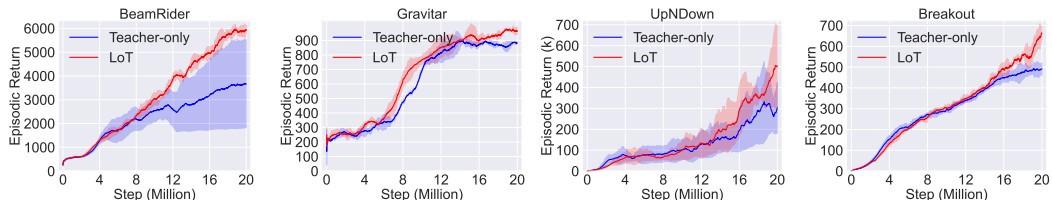

Figure 2: The episodic return of the teacher agent in LoT and the Teacher-only on four Atari games (averaged over ten runs). LoT demonstrates return gains over Teacher-only on all games.

- Given the same computational budget, the sophisticated students achieve lower final KL losses on both the training and test sets compared to the deceptive students. This suggests that the student can more effectively imitate the prediction distribution of a teacher that captures generalizable correlations.

- The deceptive students require more training steps to achieve the same training and test student-teacher KL losses as the sophisticated students. This indicates that learners tend to grasp spurious correlations much more slowly than generalizable correlations.

These results suggest that generalizable correlations are easier to imitate than spurious ones. In LoT, we expect the teacher model to master generalizable correlations by incorporating feedback from students via the LoT regularizer.

### 3.2 Atari Games

We conduct experiments on four Atari games, namely BeamRider, Breakout, UpNDown, and Gravitar, following the implementation in Huang et al. [42]. Both the LoT and Teacher-only agents have identical hyperparameters. All agents are trained using Proximal Policy Optimization (PPO) [83]. While the teacher agents interact with the game environment, the student agents are trained on the **most recent** 10,240 **samples** generated by the teacher agents, ensuring that LoT and Teacher-only experience the same environmental interactions. We use different $\alpha$ values for various games and set $N = 5$ to efficiently imitate the teacher. More details are provided in Appendix D.

The empirical results are presented in Figure 2, and we make the following observations:

- LoT improves the agent return compared to the Teacher-only version with 20 million teacher training steps. Specifically, LoT achieves $\{63.14\%, 9.79\%, 66.48\%, 35.70\%\}$ normalized return enhancements on {BeamRider, Gravitar, UpNDown, Breakout}.

- The performance gain of LoT becomes more prominent as the training progresses (from 15 million to 20 million steps).

These results suggest that LoT is an effective approach for enhancing the generalization of RL agents, as it requires no additional environmental interactions while delivering significant performance gains.

### 3.3 Language Modeling

Language modeling is a widely acknowledged NLP task, and regularization techniques have been demonstrated to significantly enhance performance in this domain [106]. To examine the impact of LoT on language modeling, we conduct experiments in two scenarios: unsupervised language pretraining and supervised fine-tuning.

#### 3.3.1 Unsupervised Language Pretraining

We conduct experiments of LoT and Teacher-only using LSTM [39], AWD-LSTM [66], and Transformer-XL [22] for teacher and student on Penn Tree Bank (PTB) [64] and WikiText-103 [65]. We follow the implementations outlined in Dai et al. [22], Merity et al. [66], Zaremba et al. [106]. In LoT, we utilize different coefficients $\alpha$ for various architectures and benchmarks to control the LoT regularizer. To ensure a fair comparison, we maintain the same total number of training steps

Table 1: The test perplexity of the teacher model in LoT and the baseline on PTB and WikiText-103. Results are averaged over three runs. LoT achieves consistent perplexity reduction over different choices of architectures and benchmarks.

| Dataset | Teacher | Student | Teacher #Param. | Teacher-only | LoT |
|---------|---------|---------|-----------------|--------------|-----|
| PTB | LSTM | LSTM | 20M | $82.75 \pm 0.36$ | $\mathbf{71.72} \pm 0.54$ |
|  | AWD-LSTM | AWD-LSTM | 24M | $58.69 \pm 0.37$ | $\mathbf{53.31} \pm 0.56$ |
| WikiText-103 | Transformer-XL-B | Transformer-XL-B | 151M | $23.72 \pm 0.41$ | $\mathbf{21.65} \pm 0.38$ |
|  | Transformer-XL-L | Transformer-XL-L | 257M | $18.50 \pm 0.25$ | $\mathbf{16.47} \pm 0.23$ |

(with teacher and student training steps accumulated) for LoT and the Teacher-only setup. Please refer to Appendix D for more implementation details.

From the empirical results presented in Table 1, we observe that LoT achieves notable perplexity (PPL) gains across various architectures and benchmarks under the same number of learning steps as Teacher-only. Specifically, LoT achieves at least 2 points PPL gains across all settings, and a 11.03 gain for LSTM on PTB. It indicates that LoT can be effectively applied to both LSTM and Transformer architectures in language pretraining.

### 3.3.2 Supervised Fine-tuning

Furthermore, to evaluate the effectiveness of LoT in fine-tuning pretrained large language models (LLMs), we conduct supervised fine-tuning (SFT) experiments using LLaMA-1 [96] and LLaMA-2 [97] on two mathematical reasoning benchmarks: GSM8K [19] and MATH [35].

We compare LoT to in-context learning (ICL) [9] and SFT. Following Touvron et al. [97], the number of in-context examples is 8 for GSM8K and 4 for MATH. The SFT configuration follows Yue et al. [105], and we fine-tune the LLaMA models for four epochs. In LoT , the teacher and student models share the same architecture for simplicity. The models are trained for two epochs in LoT to match the total training steps in SFT for fair comparison. All other configurations are consistent with those used in SFT. More implementation details are described in Appendix D.

We measure the accuracy of greedy decoding results in Table 2, and we observe that LoT enhances reasoning abilities on all architecture and dataset choices. This indicates the competence of LoT in improving the fine-tuning performance with a computational cost comparable to SFT.

Table 2: The accuracy of the teacher model in LoT and the baseline on GSM8K and MATH. Results are averaged over three runs.

| Setting | GSM8K | MATH |
|---------|-------|------|
| LLaMA-1 7B$_{+\text{ICL}}$ | $10.69 \pm 0.87$ | $2.84 \pm 0.25$ |
| LLaMA-1 7B$_{+\text{SFT}}$ | $34.39 \pm 1.28$ | $4.78 \pm 0.23$ |
| LLaMA-1 7B$_{+\text{LoT}}$ | $\mathbf{36.42} \pm 1.46$ | $\mathbf{5.39} \pm 0.28$ |
| LLaMA-2 7B$_{+\text{ICL}}$ | $14.62 \pm 0.96$ | $2.46 \pm 0.25$ |
| LLaMA-2 7B$_{+\text{SFT}}$ | $39.81 \pm 1.34$ | $5.79 \pm 0.31$ |
| LLaMA-2 7B$_{+\text{LoT}}$ | $\mathbf{41.87} \pm 1.62$ | $\mathbf{6.28} \pm 0.22$ |

### 3.4 Image Classification

To investigate the effects of LoT on computer vision tasks, we apply LoT to image classification by conducting experiments using ResNets [34], MobileNetV2 [81], ViT [24], and Swin [61] architectures pretrained on ImageNet-1K and ImageNet-21K [23] as teacher and student models. We choose CIFAR-100 [49] and ImageNet-1K as the downstream datasets. The total training steps for LoT and the Teacher-only approach are the same for a fair comparison. Further implementation details are provided in Appendix D. We conclude the following observations from results in Table 3:

- LoT achieves accuracy gains across various architectures and datasets without additional computational costs. For example, LoT improves test accuracy by almost 2 points using a ResNet-18 teacher and a ResNet-50 student on CIFAR-100 after pretrained on ImageNet-1K. Similarly, on the larger-scaled ImageNet dataset ImageNet-21K, LoT still obtains nearly 1 point improvement using ViT-B/16 as the teacher and ViT-L/16 as the student.

- The generalization of teacher models can be effectively enhanced by students of larger sizes. For instance, ResNet-50, ViT-L/16, and Swin-L students can enhance the performance of ResNet-18, ViT-B/16, and Swin-B teachers, respectively. Similarly, small student models

can also enhance the generalization performance of larger teacher models using LoT . For example, a MobileNetV2 student improves the performance of RestNet-18 and ResNet-50 by more than 1 point on CIFAR-100 with a much smaller model size. Similar results appear on the ViT-L/16 teacher and ViT-B/16 student combination in the ImageNet-1K task.

- For transformer-based models, employing different architectures for teachers and students achieves better performance than sharing the same architecture. For example, when applying a ViT-B/16 student, a ViT-L/16 teacher achieves $0.27\%$ more accuracy than using a ViT-L/16 student. This suggests that using different architectures for teacher and student increases information diversity, which contributes to enhanced generalization for teacher models [84].

These experimental results demonstrate the effectiveness of LoT in enhancing the generalization of pretrained CNN-based and Transformer-based vision models in image classification.

## 3.5 Analysis of Computational Cost and Efficiency

For supervised and unsupervised tasks, LoT involves training teacher models alongside student models as outlined in Algorithm 1. Compared to Teacher-only, the potential limitation of LoT is that it requires additional computation and memory for the student models. Therefore, in our results in Section 3, we maintain the same total training steps between LoT (accumulated for the teacher and student) and Teacher-only and demonstrate that LoT achieves better generalization performance under the same number of updates. In this regard, we show the test accuracy of image classification between LoT and Teacher-only using ViT models with respect to the total training steps in Figure 3. We note that LoT achieves better test accuracy than Teacher-only in both ViT-B/16 and ViT-L/16 with fewer total training steps. Moreover, we demonstrate that LoT remains effective even when the student model

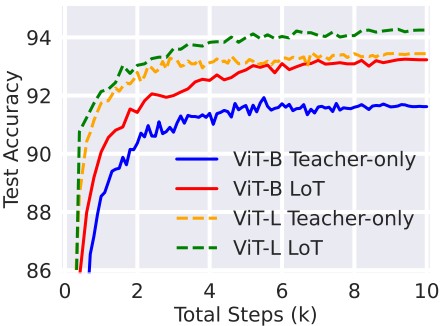

Figure 3: Test accuracy of teacher models in LoT and Teacher-only using ViT-B/16 and ViT-L/16 on CIFAR-100. LoT achieves higher test accuracy with fewer training steps.

is smaller than the teacher model in Table 3, which further reduces the computation cost compared to Teacher-only in the same total training steps and accommodates different student model choices with resource constraints. We provide more results regards efficiency of LoT in Appendix H.

In RL tasks, only the teacher model interacts with the environment to collect samples, and the student can learn from the teacher samples exclusively (please refer to Appendix G for the algorithm of

Table 3: The test accuracy of the teacher model for various teacher-student model combinations in LoT and the baseline. Results are averaged over three runs. LoT consistently enhances test performance in all model choices and datasets.

| Pretrained | Downstream | Teacher | Student | Image Size | Teacher/Student #Param. | Teacher-only | LoT |
|---|---|---|---|---|---|---|---|
| ImageNet-1K | CIFAR-100 | ResNet-18 | MobileNetV2 | $224^2$ | 12M / 4M | $81.14 \pm 0.58$ | $\mathbf{82.78} \pm 0.36$ |
| | | ResNet-18 | ResNet-18 | $224^2$ | 12M / 12M | $81.14 \pm 0.58$ | $\mathbf{82.89} \pm 0.25$ |
| | | ResNet-18 | ResNet-50 | $224^2$ | 12M / 26M | $81.14 \pm 0.58$ | $\mathbf{83.13} \pm 0.26$ |
| | | ResNet-50 | MobileNetV2 | $224^2$ | 26M / 4M | $84.09 \pm 0.32$ | $\mathbf{85.38} \pm 0.44$ |
| | | ResNet-50 | ResNet-18 | $224^2$ | 26M / 12M | $84.09 \pm 0.32$ | $\mathbf{85.77} \pm 0.19$ |
| | | ResNet-50 | ResNet-50 | $224^2$ | 26M / 26M | $84.09 \pm 0.32$ | $\mathbf{86.04} \pm 0.38$ |
| ImageNet-21K | CIFAR-100 | ViT-B/16 | ViT-B/16 | $384^2$ | 86M / 86M | $91.57 \pm 0.31$ | $\mathbf{93.17} \pm 0.35$ |
| | | ViT-B/16 | ViT-L/16 | $384^2$ | 86M / 307M | $91.57 \pm 0.31$ | $\mathbf{93.25} \pm 0.44$ |
| | | ViT-L/16 | ViT-B/16 | $384^2$ | 307M / 86M | $93.44 \pm 0.28$ | $\mathbf{94.29} \pm 0.33$ |
| | | ViT-L/16 | ViT-L/16 | $384^2$ | 307M / 307M | $93.44 \pm 0.28$ | $\mathbf{94.18} \pm 0.26$ |
| ImageNet-21K | ImageNet-1K | ViT-B/16 | ViT-B/16 | $384^2$ | 86M / 86M | $83.97 \pm 0.11$ | $\mathbf{84.54} \pm 0.15$ |
| | | ViT-B/16 | ViT-L/16 | $384^2$ | 86M / 307M | $83.97 \pm 0.11$ | $\mathbf{84.80} \pm 0.08$ |
| | | ViT-L/16 | ViT-B/16 | $384^2$ | 307M / 86M | $85.15 \pm 0.17$ | $\mathbf{85.92} \pm 0.09$ |
| | | ViT-L/16 | ViT-L/16 | $384^2$ | 307M / 307M | $85.15 \pm 0.17$ | $\mathbf{85.65} \pm 0.11$ |
| | | Swin-B | Swin-B | $384^2$ | 88M / 88M | $86.37 \pm 0.06$ | $\mathbf{86.68} \pm 0.15$ |
| | | Swin-B | Swin-L | $384^2$ | 88M / 197M | $86.37 \pm 0.06$ | $\mathbf{86.73} \pm 0.14$ |
| | | Swin-L | Swin-B | $384^2$ | 197M / 88M | $87.27 \pm 0.11$ | $\mathbf{87.64} \pm 0.12$ |
| | | Swin-L | Swin-L | $384^2$ | 197M / 197M | $87.27 \pm 0.11$ | $\mathbf{87.59} \pm 0.09$ |

Table 4: Performance comparison of Teacher-only, BAN and LOT on CIFAR-100. LOT achieves superior performance to Teacher-only and BAN.

| Dataset | Teacher | Student | Teacher-only | BAN (Student) | LOT (Teacher) |
|---|---|---|---|---|---|
| CIFAR-100 | ResNet-18 | ResNet-18 | 81.14 | 82.08 | **82.89** |
| CIFAR-100 | ResNet-50 | ResNet-50 | 84.09 | 84.73 | **86.04** |
| CIFAR-100 | ViT-B/16 | ViT-B/16 | 91.57 | 92.44 | **93.17** |
| CIFAR-100 | ViT-L/16 | ViT-L/16 | 93.44 | 93.82 | **94.18** |

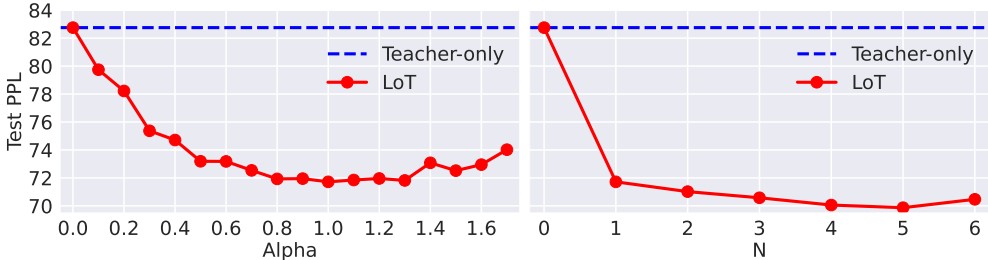

Figure 4: Effects of regularization coefficient $\alpha$ (left) and student steps ratio $N$ (right). $\alpha = 1$ is the best $\alpha$ value to achieve the lowest test perplexity of the teacher model, and moderate student steps ratio $N$ such as 4 and 5 benefit the teacher model the most.

PPO-version LOT ). Therefore, LOT introduces negligible computation costs since sample collections are more resource-intensive than fitting the agent network to the samples in RL. For instance, in our Atari games experiments, the training time of LOT (606 minutes) is comparable to the Teacher-only setting (597 minutes) on a single NVIDIA A6000 GPU.

### 3.6 Additional Investigation

**Comparison to KD.** To investigate the effect of LOT compared to other student-teacher learning paradigms, we compare LOT to the born-again networks (BAN) baseline [29]. In BAN, we select the checkpoint with the best performance of the Teacher-only model as the (frozen) teacher and distill its knowledge into a student model with an identical architecture. Equal weights are assigned to the hard loss (from the dataset) and soft loss (from the teacher) to train the student model [36]. All other configurations remain consistent with LOT . The results in Table 4 indicate that LOT achieves superior performance than BAN with a strong feedback model, further indicating the significance of the interactive learning process in LOT .

**Effect of regularization coefficient $\alpha$.** The strength of regularization plays a crucial role in the overall training effect [50]. To investigate the effects of LOT on the generalization of the teacher model, we perform experiments on PTB using the LSTM architecture for both teacher and student models. The configuration follows Section 3.3, except that we gradually increase the value of $\alpha$ in LOT from 0 to 1.7 and examine the test PPL of the teacher model. The results are presented in Figure 4 (left). We observe that the performance of the teacher model improves rapidly as $\alpha$ increases from 0 to 1, and when the value exceeds this point, the performance of the teacher begins to decline. This observation suggests that moderate feedback from the student is most beneficial for the teacher, but an excessively strong signal can hinder the teacher's learning process. Similar effects of large $\alpha$ values have been noted in joint teacher-student training in knowledge distillation [75].

**Effect of student steps ratio $N$.** To demonstrate the importance of the student steps ratio $N$ in LOT , we conduct additional experiments by training LSTM teacher and student models on PTB using various values of $N$. The empirical results presented in Figure 4 (right) indicate that the teacher benefits most from a moderate $N$ value, such as 4 or 5. This finding suggests that achieving a balanced ratio between teacher and student model updates is crucial for optimal performance. When $N$ is too low, the student may not sufficiently learn from the teacher, thereby reducing the quality of the feedback it provides. Conversely, if $N$ is too high, the student may overfit the teacher's errors, resulting in less effective imitability measurement.

# 4   Conclusion

Identifying generalizable multiscale correlations from the vast space of possible correlations remains a significant challenge in machine learning. Inspired by cognitive science beliefs about human intelligence, we have shown experimentally that generalizable correlations are more imitable by other learners. In particular, we introduced a novel regularization method, LOT , which identifies generalizable correlations by teaching student models and exploiting their feedback. We conducted comprehensive experiments across various learning tasks and neural architectures. The results demonstrate that our proposed regularizer enhances model performance effectively and efficiently. In conclusion, our proposed LOT regularization offers a promising new approach to improve the generalization of neural networks by leveraging the learning process of student models and incorporating their feedback to refine the teacher model.

# 5   Acknowledgments

Metaxas is partially supported by research grants from NSF: 2310966, 2235405, 2212301, 2003874, 1951890, AFOSR 23RT0630, and NIH 2R01HL127661.

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

# A  Ethics and Social Impacts

In this work, we propose a regularization method to improve the generalization of deep neural networks. Our work focuses on technical contributions to deep learning and AI. Therefore, the potential social impacts of AI in general apply to our work, including fake information, toxic content, fairness concerns, and misuse of AI. For example, toxic content like hate speech can lead to data contamination and therefore have harmful impacts on society, which has been observed in large-scale pretrained models. By employing our method, such harmful behavior can potentially be amplified.

# B  Related Works

## B.1  Regularization in Deep Learning

Regularization serves as a primary strategy to improve generalization capabilities and mitigate over-fitting [52]. Various regularization techniques exist for deep neural networks. One of the earliest and most straightforward approaches to regularization involves constraining the model's capacity by adding a penalty function to the original objective function. Techniques such as L1 regularization [41, 92, 93], L2 regularization [20, 50, 82], and weight decay [50, 104] fall into this category. Introducing noise [38, 77] to the system can also judiciously enhance generalizability and prevent over-fitting. Dropout [4, 37, 88, 100] is a widely used regularization technique that randomly drops certain neural network connections during training.

## B.2  Student-Teacher Learning Paradigms

### B.2.1  Knowledge Distillation

Knowledge distillation (KD) is a technique that transfers knowledge from a teacher model to a student model by training the student to imitate the teacher's outputs [36]. This approach is widely applied in areas such as model compression, transparency, and interpretability [8, 10, 27, 36, 60, 91]. Model compression is often motivated by resource constraints. Pioneering works include Buciluǎ et al. [10], which compresses ensemble neural networks into a single network, and Ba and Caruana [3], which improves shallow neural network accuracy by mimicking deep networks. KD is also applied in various domains, including deep reinforcement learning [80], continual learning [28, 59, 85], and learning privileged information theory [62, 76]. The dark knowledge method [36] further develops KD, where a student model aims to fully match the output distribution of the teacher. Intuitively, distillation is effective because the teacher's output distribution over classes provides a more informative training signal than a one-hot label. Additionally, in born-again networks (BAN) [29], the teacher and student have identical neural architecture and model sizes, but the student can surprisingly surpass the teacher's accuracy.

### B.2.2  Language Emergence

In a cooperative environment, agents can learn emergent languages for communication to solve specific tasks. The emergence of such communication protocols is extensively studied in the context of multi-agent referential games [26, 55]. In these games, one agent is required to describe its observations to another agent, which is then tasked with deducing the initial agent's observations [53]. The majority of methods employed to learn discrete communication protocols between agents utilize RL [26, 99]. Compositionality is a desirable feature in the language used by agents, as it facilitates flawless generalization for previously unseen combinations of attributes [5, 11, 17, 48]. However, the community still lacks strong research indicating what general conditions are necessary or sufficient for compositional language emergence. Chaabouni et al. [11, 12], Galke et al. [30], Li and Bowling [56] postulate that compositional languages are more straightforward to learn.

# C  Motivation and Insights of Our Method

The concept of Learning from Teaching originates in cognitive psychology and linguistics, particularly within the iterated learning theory of language emergence [45–47, 86]. This theory posits that the generalizable nature of languages arises from the iterative learning process across generations in a

society. The core hypothesis is that a generalizable language is inherently easier to teach and learn [56, 78, 98], which aligns with our main hypothesis.

In the AI community, recent research has aimed to employ iterated learning to enhance the generalization of emergent languages and language acquisition among artificial learners. For example, some studies have used iterated learning to improve the generalization of emergent languages between AI agents [56, 78], while others have applied it to address generalization challenges in tasks like compositional Visual Question Answering (VQA) [98]. LoT shares the same motivation as this line of research. Our primary contribution extends the concept of "ease-of-teaching" [56] from language learning to a broader range of machine learning tasks, including supervised, unsupervised learning, and reinforcement learning.

LoT functions as a regularizer, similar to other commonly used regularizers like the L2 regularizer. The L2 regularizer is effective because it encourages neural networks to learn simpler correlations, thereby avoiding overfitting. It is widely accepted that correlations with lower Kolmogorov complexity are more generalizable if they can perfectly explain a complex dataset. This aligns with the idea that "generalization equals optimal compression," as discussed by Ilya Sutskever [89]. Essentially, this notion adapts Occam's Razor to the field of AI. Our key insight is that the "ease-of-teaching" metric serves as an effective regularizer beyond language emergence tasks.

Consider an intuitive example: Student A learns math by rote memorization, while Student B understands the core concepts and only memorizes essential rules, deducing the rest when needed. Both approaches can perform similarly on simple problem sets. However, as data complexity increases, Student A's burden grows significantly, while Student B's understanding-based approach remains manageable. Consequently, Student B's knowledge is easier to teach to another student, as it involves less complexity. Therefore, teachability (or imitability) can serve as a proxy for complexity.

## D   Implementation Details.

**Atari Games.**   We perform experiments on four Atari games, namely Beam-Rider, Breakout, UpNDown, and Gravitar, following the implementation outlined in [42]. We set the regularization coefficient $\alpha$ to $0.5$ for BeamRider, Breakout, and UpNDown, and to $0.1$ for Gravitar. The other hyperparameters remain consistent across all four games. We use $N$ of $5$. For all agents, the optimizer employed is Adam, with an initial learning rate of $0.00025$. The teacher agent is trained for a total of $20,000,000$ timesteps. The temperature used in the KL loss is set to $1$. The experiments are implemented on the NVIDIA A6000 48GB GPUs.

**Language Modeling.**   In the training-from-scratch experiments, we use the Transformer-XL architecture following Dai et al. [22], the LSTM architecture following Zaremba et al. [106], and the AWD-LSTM architecture following Merity et al. [66]. For supervised fine-tuning experiments with LLaMA-1 and LLaMA-2, we employ the hyperparameters described in Yue et al. [105] and use the HuggingFace Transformers library [101]. The hyperparameters for LoT are detailed in Table 5. The experiments for LSTM and AWD-LSTM are implemented on one single NVIDIA A100 40GB GPU. The Transformer-XL and LLaMA of LoT are trained on 4 and 8 NVIDIA A100 40GB GPUs, respectively.

| Model | Dataset | $\alpha$ | $N$ | Optimizer | Learning Rate | Training Epochs/Steps | Temperature |
|---|---|---|---|---|---|---|---|
| LSTM | PTB | 1.0 | 1 | SGD | 30 | 30 Epochs | 1.5 |
| AWD-LSTM | PTB | 1.0 | 1 | ASGD | 30 | 250 Epochs | 1.5 |
| Transformer-XL-B | WikiText-103 | 0.1 | 1 | ADAM | 0.01 | 60,000 Steps | 2 |
| Transformer-XL-L | WikiText-103 | 0.1 | 1 | ADAM | 0.01 | 150,000 Steps | 2 |
| LLaMA-1 7B | GSM8K | 0.01 | 1 | ADAMW | $2 \times 10^{-5}$ | 2 Epochs | 2 |
| LLaMA-1 7B | MATH | 0.01 | 1 | ADAMW | $2 \times 10^{-5}$ | 2 Epochs | 2 |
| LLaMA-2 7B | GSM8K | 0.01 | 1 | ADAMW | $2 \times 10^{-5}$ | 2 Epochs | 2 |
| LLaMA-2 7B | MATH | 0.01 | 1 | ADAMW | $2 \times 10^{-5}$ | 2 Epochs | 2 |

Table 5: Hyperparameters for Language Modeling.

**Image Classification.**   For CNN experiments, we use the ImageNet-1K pretrained architectures MobileNetV2 and ResNets, which can be downloaded from the official PyTorch Model Zoo[2]. For

---

[2]`https://pytorch.org/vision/stable/models.html`

ViT and Swin experiments, we follow the implementations described in Dosovitskiy et al. [24] and Liu et al. [61], using the official ImageNet-1K or ImageNet-21K pretrained weights downloaded from [3] and [4]. The optimal hyperparameters for LoT are obtained through grid research. The detailed hyperparameters are illustrated in Table 6. The experiments for MobileNetV2 and ResNets are implemented on one single NVIDIA A100 40GB GPU. The ViT and Swin experiments are implemented on 4 NVIDIA A100 40GB GPUs.

| Model | Dataset | $\alpha$ | $N$ | Optimizer | Learning Rate | Training Epochs/Steps | Temperature |
|-------|---------|----------|-----|-----------|---------------|----------------------|-------------|
| MobileNetV2 | CIFAR-100 | 1.0 | 1 | SGD | 0.02 | 30 Epochs | 1.5 |
| ResNet-18 | CIFAR-100 | 1.0 | 1 | SGD | 0.02 | 30 Epochs | 1.5 |
| ResNet-50 | CIFAR-100 | 1.0 | 1 | SGD | 0.02 | 30 Epochs | 1.5 |
| ViT-B/16 | CIFAR-100 | 1.0 | 1 | SGD | 0.02 | 5,000 Steps | 1.5 |
| ViT-L/16 | CIFAR-100 | 1.0 | 1 | SGD | 0.02 | 5,000 Steps | 1.5 |
| ViT-B/16 | ImageNet-1K | 1.0 | 1 | SGD | 0.03 | 10,000 Steps | 1.5 |
| ViT-L/16 | ImageNet-1K | 1.0 | 1 | SGD | 0.03 | 10,000 Steps | 1.5 |
| Swin-B | ImageNet-1K | 0.5 | 1 | ADAMW | $2 \times 10^{-5}$ | 15 Epochs | 1.5 |
| Swin-L | ImageNet-1K | 0.5 | 1 | ADAMW | $2 \times 10^{-5}$ | 15 Epochs | 1.5 |

Table 6: Hyperparameters for Image Classification.

## E    Scalability Analysis

From our extensive results shown in Section 3, LoT proves to be widely applicable across various domains, including reinforcement learning (Section 3.2), unsupervised learning (Section 3.3), and supervised learning (Section 3.4). It can be effectively applied to different architectures such as CNN-based (Table 3), LSTM-based (Table 1), and Transformer-based(Table 1) models. LoT works well on both small datasets like PTB (Table 1) and CIFAR-100 (Table 3), and large datasets such as WikiText-103 (Table 1) and ImageNet (Table 3). It is also suitable for both small models like ResNets (Table 3) and large models like ViT (Table 3) and LLaMA (Table 2). Additionally, LoT is compatible with existing regularization methods such as weight decay and dropout. In our experiments with ResNets, weight decay was applied to both LoT and Teacher-only setups. In the experiments with Transformer-XL, ViT, and Swin, dropout is applied to both LoT and Teacher-only setups.

## F    Limitation

A potential limitation of LoT lies in the additional computational and memory costs required for training the student models. However, as demonstrated in Section 3.5, LoT achieves better generalization with fewer training steps compared to Teacher-only models, and the flexibility in choosing student models can accommodate varying resource constraints. In RL, the additional computational costs introduced by LoT are negligible, as sample collection is more resource-intensive than fitting the agent networks to the samples, as discussed in Section 3.5. Moreover, in real-world settings, inference cost is more critical than training cost. The superior generalization achieved by LoT offers significant benefits during inference without introducing additional inference costs.

## G    Algorithm for the PPO-version of Our Method

The LoT algorithm for Proximal Policy Optimization (PPO) is illustrated in Algorithm 2. In our experiments, the teacher's sampled data $\mathcal{B}_t$ is continuously added to the student sample collections $\mathcal{D}_s$. Meanwhile, the most recent samples from $\mathcal{D}_s$ are used to formulate the student training batch $\mathcal{B}_s$ to ensure a high quality of its training dataset.

## H    Additional Results

**Computational Efficiency.**    To further demonstrate the computational efficiency and superiority of LoT , we conduct experiments using LSTM on PTB and ViT-B/16 on CIFAR-100 with varying training epochs and steps, while keeping other configurations the same as in Section 3.3 and Section 3.4. The results presented in Table 7 demonstrate that given equivalent computational budgets, LoT consistently outperforms the Teacher-only model across various datasets and architectures, even when the Teacher-only model trains for twice the number of epochs and steps. This further highlights

---

[3]`https://github.com/google-research/vision_transformer`
[4]`https://github.com/microsoft/Swin-Transformer`

**Algorithm 2** Learning from Teaching for PPO

1: **Input:** Regularization Coefficient $\alpha > 0$, Student Steps Ratio $N > 0$.
2: Initialize teacher network $T$ parameterized by $\boldsymbol{\theta}$ and student networks $S_i, i = 1, 2, \cdots K$, parameterized by $\boldsymbol{\phi}$.
3: Initialize replay buffer $\mathcal{D}_s = \emptyset$
4: **repeat**
5:      Sample minibatch $\mathcal{B}_t$ by running $T$ in simulator, add $\mathcal{B}_t$ to $\mathcal{D}_s$
6:      Sample a batch of data $\mathcal{B}_s \subset \mathcal{D}_s$
7:      Compute $\tilde{R}(\boldsymbol{\theta}) = \frac{\alpha}{|\mathcal{B}_s|} \sum_{\mathbf{x} \in \mathcal{B}_s} \sum_{i=1}^{K} \lambda_i \mu_{t,s_i}(\mathbf{x})$
8:      Compute $\tilde{L}_t(\boldsymbol{\theta})$ using the PPO loss on minibatch $\mathcal{B}_t$
9:      Update $\boldsymbol{\theta}$ using gradient $\nabla_{\boldsymbol{\theta}} \tilde{L}_t(\boldsymbol{\theta})$
10:      Fit value network for PPO on minibatch $\mathcal{B}_t$
11:      **for** $i = 1$ **to** $N$ **do**
12:          Sample $\mathcal{B}_s \subset \mathcal{D}_s$
13:          Compute $\tilde{L}_s(\boldsymbol{\phi}) = \frac{1}{|\mathcal{B}_s|} \sum_{\mathbf{x} \in \mathcal{B}_s} \sum_{i=1}^{K} \mu_{s_i,t}(\mathbf{x})$
14:          Update student networks' parameters $\boldsymbol{\phi}$ using loss gradient $\nabla_{\boldsymbol{\phi}} \tilde{L}_s(\boldsymbol{\phi})$
15:      **end for**
16: **until** $T$ converges

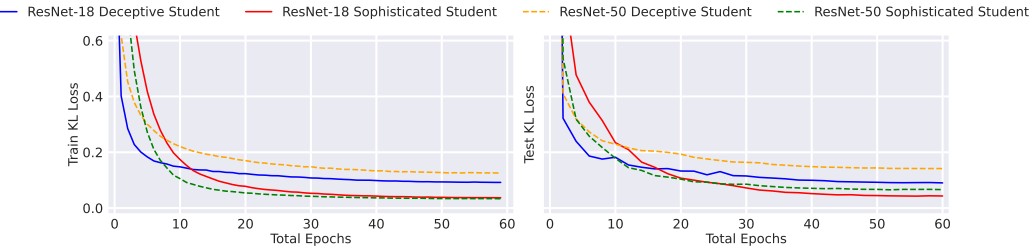

Figure 5: Training and test KL-divergence losses of student models in LoT using ResNet-18 and ResNet-50 on CIFAR-100 with different teacher models.

LoT 's effectiveness in improving the teacher model's generalization while maintaining enhanced computational efficiency.

Table 7: Performance of the teacher model in LoT and Teacher-only on image classification. The hyperparameters are the same as the corresponding experiments in the paper.

| Dataset | Teacher | Student | Total Train Epochs/Steps | Teacher-only | LoT |
|---------|---------|---------|--------------------------|--------------|------|
| CIFAR-100 | ViT-B/16 | ViT-B/16 | 10,000 steps | 91.57 | **93.17** |
| CIFAR-100 | ViT-B/16 | ViT-B/16 | 15,000 steps | 91.74 | **93.23** |
| CIFAR-100 | ViT-B/16 | ViT-B/16 | 20,000 steps | 91.82 | **93.40** |
| PTB | LSTM | LSTM | 60 epochs | 82.75 | **71.72** |
| PTB | LSTM | LSTM | 90 epochs | 82.48 | **71.22** |
| PTB | LSTM | LSTM | 120 epochs | 82.42 | **70.67** |

**Additional Evidence for Hypothesis.** We provide additional experimental results to validate our hypothesis using ResNet-50 and ResNet-18 as both the teacher and student models on CIFAR-100, following the same methodology described in Section 3.1, but with different model architectures. The training and test KL-divergence of the sophisticated and deceptive students are shown in Figure 5. We observe that the sophisticated students achieve lower final KL losses compared to the deceptive students with fewer training epochs, which further supports our hypothesis

**Out-of-distribution Performance.** We conduct additional experiments by fine-tuning models on ImageNet-1K and evaluating them on ImageNet-R and ImageNet-Sketch using ViT-B/16 and ViT-L/16 models to investigate the out-of-distribution robustness of LoT . The results, shown in

Table 8, demonstrate that LoT also brings performance improvements on these datasets, indicating the robustness of LoT across a broader set of scenarios.

**Additional Comparison to KD.** In Table 4, we show that LoT outperforms the distillation method BAN. To provide stronger validation of LoT 's effectiveness, we conduct additional experiments using ResNet-50 and ViT-B/16 on CIFAR-100. We compare LoT to distillation methods such as BAN, DKD [108], and ReviewKD [15], with the teacher weights in these methods being the best checkpoint of Teacher-only. The results, shown in Table 9, indicate that LoT achieves better performance than these distillation baselines, further underscoring the effectiveness of the unique interactive learning process of LoT .

Table 8: Performance of LoT and Teacher-only on ImageNet-R and ImageNet-Sketch.

| Dataset | Teacher | Student | Teacher-only / LoT |
|---|---|---|---|
| ImageNet-R | ViT-B/16 | ViT-B/16 | 49.11 / 52.27 |
| ImageNet-R | ViT-B/16 | ViT-L/16 | 49.11 / 54.08 |
| ImageNet-R | ViT-L/16 | ViT-B/16 | 54.42 / 58.18 |
| ImageNet-R | ViT-L/16 | ViT-L/16 | 54.42 / 57.79 |
| ImageNet-Sketch | ViT-B/16 | ViT-B/16 | 38.85 / 41.46 |
| ImageNet-Sketch | ViT-B/16 | ViT-L/16 | 38.85 / 42.89 |
| ImageNet-Sketch | ViT-L/16 | ViT-B/16 | 43.83 / 47.61 |
| ImageNet-Sketch | ViT-L/16 | ViT-L/16 | 43.83 / 45.91 |

Table 9: Performance of LoT , BAN, ReviewKD, DKD on CIFAR100.

| Method | Teacher | Student | Accuracy |
|---|---|---|---|
| Teacher-only | ResNet-50 | N/A | 84.09 |
| BAN | ResNet-50 | ResNet-50 | 84.73 |
| ReviewKD | ResNet-50 | ResNet-50 | 85.31 |
| DKD | ResNet-50 | ResNet-50 | 85.17 |
| LoT | ResNet-50 | ResNet-50 | 86.04 |
| Teacher-only | ViT-B/16 | N/A | 91.57 |
| BAN | ViT-B/16 | ViT-B/16 | 92.44 |
| ReviewKD | ViT-B/16 | ViT-B/16 | 92.73 |
| DKD | ViT-B/16 | ViT-B/16 | 92.82 |
| LoT | ViT-B/16 | ViT-B/16 | 93.17 |

**Results on Validation Datasets.** We provide additional results on the official validation datasets for PTB and WikiText-103 in Table 10. These results demonstrate that LoT consistently outperforms the Teacher-only approach on both the validation and test datasets for PTB and WikiText-103, further validating the effectiveness of LoT .

Table 10: Test/Validation perplexity of LoT and Teacher-only on the official test/validation datasets.

| Dataset | Teacher | Student | Teacher-only (Valid) | Teacher-only (Test) | LoT (Valid) | LoT (Test) |
|---|---|---|---|---|---|---|
| PTB | LSTM | LSTM | 86.02 | 82.75 | 73.98 | 71.72 |
| PTB | AWD-LSTM | AWD-LSTM | 60.62 | 58.69 | 55.07 | 53.31 |
| Wikitext-103 | Transformer-XL-B | Transformer-XL-B | 24.68 | 23.72 | 22.24 | 21.65 |
| Wikitext-103 | Transformer-XL-L | Transformer-XL-L | 18.65 | 18.50 | 16.41 | 16.47 |

**Performance of Student Models.** We present the results for the student models in Table 11. Our observations indicate that when the student and teacher models share the same architecture, the student models can achieve performance levels comparable to those of the teacher models. While the performance of the student models improves under LoT , it is important to highlight that LoT is primarily designed to enhance the generalization capabilities of the teacher model.

**Detailed Computation Cost.** We provide a detailed comparison of the computational budget for LoT and Teacher-only in Table 12. Our analysis shows that LoT uses the same number of CPU cores as Teacher-only, with GPU usage being 12% to 55% higher. Despite this, LoT exhibits lower training times compared to Teacher-only (except in RL tasks) when subjected to the same total training epochs/steps, while still achieving significant performance improvements.

Table 11: The performance of student models in LoT on language modeling and image classification.

| Task | Dataset | Teacher | Student | Teacher-only | LoT (Teacher) | LoT (Student) |
|---|---|---|---|---|---|---|
| Language Modeling | PTB | LSTM | LSTM | 82.75 | 71.72 | 73.33 |
| Language Modeling | WikiText-103 | Transformer-XL-L | Transformer-XL-L | 18.50 | 16.47 | 16.89 |
| Image Classification | CIFAR100 | ResNet-50 | ResNet-18 | 84.09 | 85.77 | 83.24 |
| Image Classification | CIFAR100 | ResNet-50 | ResNet-50 | 84.09 | 86.04 | 85.72 |
| Image Classification | ImageNet-1K | ViT-B/16 | ViT-B/16 | 91.57 | 93.17 | 92.95 |
| Image Classification | ImageNet-1K | ViT-B/16 | ViT-L/16 | 91.57 | 93.25 | 93.89 |

Table 12: Computational resources, memory usage, and training time of LoT and Teacher-only.

| Dataset | Teacher Model / Student Model | Total Train Steps (teacher+student) | Computational Resources | CPU Usage (Teacher-only/LoT) | GPU Usage (Teacher-only/LoT) | Training Time (Teacher-only/LoT) | Performance (Teacher-only/LoT) |
|---|---|---|---|---|---|---|---|
| BeamRider | Standard Network / Standard Network | 20M frames | 1 NVIDIA A6000 48GB GPU | 16 core / 16 core | 0.8 GB / 0.9 GB | 10 h / 10.1 h | 3,651 score / 5,956 score (↑) |
| PTB | LSTM / LSTM | 60 epochs | 1 × NVIDIA A100 40GB GPU | 1 core / 1 core | 1.1 GB / 1.5 GB | 0.6 h / 0.3 h | 82.8 ppl / 71.7 ppl (↓) |
| WikiText-103 | Transformer-XL-L / Transformer-XL-L | 0.3M steps | 4 × NVIDIA A100 40GB GPU | 4 core / 4 core | 4 × 21.4 GB / 4 × 33.2 GB | 85.6 h / 67.7 h | 18.5 ppl / 16.5 ppl (↓) |
| GSM8K | LLaMA-2 7B / LLaMA-2 7B | 4 epochs | 8 × NVIDIA A100 40GB GPU | 8 core / 8 core | 8 × 27.4 GB / 8 × 39.8 GB | 8.1 h / 6.7 h | 39.8 acc / 41.9 acc (↑) |
| CIFAR100 | ResNet-50 / ResNet-18 | 60 epochs | 1 × NVIDIA A100 40GB GPU | 1 core / 1 core | 13.6 GB / 16.7 GB | 0.7 h / 0.5 h | 84.1 acc / 85.8 acc (↑) |
| ImageNet-1K | ViT-L/16 / ViT-B/16 | 20K steps | 4 × NVIDIA A100 40GB GPU | 4 core / 4 core | 4 × 17.5 GB / 4 × 23.1 GB | 28.9 h / 18.7 h | 85.2 acc / 86.0 acc (↑) |

**Ablation of Metrics in LoT Regularizer.**  We conduct experiments with different metrics for the "imitability" measurement, such as L2 loss. However, we find that using KL-divergence achieves better performance compared to L2 loss. The results of utilizing L2 loss for the LoT regularizer with ViT-B/16 and ViT-L/16 on CIFAR-100 are presented in Table 13. These results show that using L2 loss for the LoT regularizer also brings performance improvements, further indicating the effectiveness of LoT regularization.

Table 13: Performance of using L2 loss for the LoT regularizer on CIFAR100.

| Dataset | Teacher | Student | Teacher-only | LoT (KL-Divergence) | LoT (L2) |
|---|---|---|---|---|---|
| CIFAR100 | ViT-B/16 | ViT-B/16 | 91.57 | 93.17 | 92.77 |
| CIFAR100 | ViT-B/16 | ViT-L/16 | 91.57 | 93.25 | 92.94 |
| CIFAR100 | ViT-L/16 | ViT-B/16 | 93.44 | 94.29 | 94.12 |
| CIFAR100 | ViT-L/16 | ViT-L/16 | 93.44 | 94.18 | 94.05 |

