# OpenReview forum: "Learning from Teaching Regularization: Generalizable Correlations Should be Easy to Imitate"
_NeurIPS.cc/2024/Conference — NeurIPS 2024 poster_

### Official Review · Reviewer_DocJ · 2024-07-05

**Soundness:** 3
**Presentation:** 3
**Contribution:** 2
**Rating:** 6
**Confidence:** 3

**Summary:**

The authors proposed a novel regularization approach called Learning from Teaching (LoT) to enhance generalization. The hypothesis that simple correlations are generalizable is the main question for this work. Through a teacher student approach, the authors are capable of provide to the main model more generalizable and imitable

**Strengths:**

- The LoT computation measurement of ‘imitability’ through the student, later used as the regularizer, is the paper main contribution.
- Comprehensive experimental section with a broad set of experiments, such as RL, Fine-tuning and Image classification.

**Weaknesses:**

- I would like to see further evidences of the proposed hypothesis.

**Questions:**

- The hypothesis on Section 2.1 have been proposed before? I better discussion on that will help the overall reading and to better position the paper when compared to other approaches.
- For all the experiments N (student steps ratio) was 1? If not the authors compared LoT with other methods for the same number of steps? If not the superior number of steps can partially be the cause of the performance gains.
- Regarding the choice of metrics the authors chose KL-divergence, did the authors experimented with other losses?
- Not sure if it is feasible but benchmark LoT on ImageNet-R [1] and ImageNet-Sketch [2] will be interesting for a broader set of results.

[1] Hendrycks, D., Basart, S., Mu, N., Kadavath, S., Wang, F., Dorundo, E., Desai, R., Zhu, T., Parajuli, S., Guo, M., et al. The many faces of robustness: A critical analysis of out-of-distribution generalization. In CVPR pp. 8340–8349, 2021.

[2] Wang, H., Ge, S., Lipton, Z., and Xing, E. P. Learning ro- bust global representations by penalizing local predictive power. In NeurIPS, pp. 10506–10518, 2019.

**Limitations:**

- The current set of baselines can be expanded. Methods like [3] and [4] can provide a better understanding on how well LoT is positioned.

[3] Pengguang Chen, Shu Liu, Hengshuang Zhao, and Jiaya Jia. Distilling knowledge via knowledge review. In CVPR, 2021

[4]  Borui Zhao, Quan Cui, Renjie Song, Yiyu Qiu, Jiajun Liang, Decoupled Knowledge Distillation, CVPR, 2022.

---

> ### Author Rebuttal · Authors · 2024-08-07
>
> Thank you for acknowledging the novelty of our method and the comprehensiveness of our experiments.
>
> > W1: further evidences of the proposed hypothesis.
>
> Thank you for your request for further evidence supporting our hypothesis.
>
> We have provided additional experimental results to validate our hypothesis using ResNet-50 and ResNet-18 as both the teacher and student models on CIFAR-100, following the same methodology described in Section 3.1, but with different model architectures. The training and test KL-divergence of the sophisticated and deceptive students are shown in Figure 5 of the PDF in the General Responses. We observe that the sophisticated students achieve lower final KL losses compared to the deceptive students with fewer training epochs, which further supports our hypothesis.
>
> Due to the time constraints of the rebuttal process, we have included additional results for two models on CIFAR-100. We promise to provide more extensive results in the final version of our paper.
>
> > Q1: The hypothesis on Section 2.1 have been proposed before? I better discussion on that will help the overall reading and to better position the paper when compared to other approaches.
>
> Thank you for your insightful feedback. We agree that providing more context and discussion on the motivations and background of our hypothesis is crucial for better positioning our paper.
>
> For in-depth discussions, please refer to [GR1] and [GR2].
>
> The hypothesis presented in Section 2.1, in the context of human language acquisition and emergence, is a widely accepted concept in cognitive sciences and linguistics [17, 18, 19, 20]. In the AI community, similar hypotheses have been proposed concerning artificial language emergence [16]. However, to the best of our knowledge, we are the first to propose that this principle is widely applicable across different domains and can be used to deduce a practical regularizer.
>
> We will incorporate these discussions to better position our paper and provide a clearer comparison with other approaches.
>
> > Q2: For all the experiments N (student steps ratio) was 1? If not the authors compared LoT with other methods for the same number of steps? If not the superior number of steps can partially be the cause of the performance gains.
>
> Thank you for your question.
>
> Yes, in all our experiments \(N = 1\), ensuring that the total training steps (Teacher + Student) for LoT are the same as for the Teacher-only approach. This setup maintains an equal training steps budget for a fair comparison. Therefore, the performance gains observed are not due to additional training steps, as both LoT and Teacher-only utilize the same total number of steps.
>
> Additionally, we present results in Table 7 of our paper where we further increased the training steps for both LoT and Teacher-only. These results show that further increasing the training steps does not significantly improve performance, which further underscores the effectiveness of LoT in achieving performance gains under the same training steps as the Teacher-only approach.
>
> > Q3: Regarding the choice of metrics the authors chose KL-divergence, did the authors experimented with other losses?
>
> Thank you for your question regarding the choice of metrics.
>
> We have experimented with different metrics for the “imitability” measurement, such as L2 loss. However, we found that using KL-divergence achieves better performance compared to L2 loss. The results of utilizing L2 loss for the LoT regularizer with ViT-B/16 and ViT-L/16 on CIFAR-100 are presented in Table 13 in the PDF. These results show that using L2 loss for the LoT regularizer also brings performance improvements, further indicating the effectiveness of LoT regularization.
>
> > Q4: LoT on ImageNet-R [1] and ImageNet-Sketch [2] will be interesting for a broader set of results.
>
> Thank you for your suggestion.
>
> We conducted additional experiments by fine-tuning models on ImageNet-1K and evaluating them on ImageNet-R and ImageNet-Sketch using ViT-B/16 and ViT-L/16 models to investigate the out-of-distribution robustness of LoT. The results, shown in Table 8 of the PDF, demonstrate that LoT also brings performance improvements to these datasets, indicating the robustness of LoT across a broader set of scenarios.
>
> > Q5: The current set of baselines can be expanded. Methods like [3] and [4] can provide a better understanding on how well LoT is positioned.
>
> Thank you for your suggestion regarding expanding the set of baselines.
>
> In our paper, we have already demonstrated that LoT achieves better performance than the distillation method BAN [23] (Table 4). To further validate LoT’s effectiveness, we conducted additional experiments using ResNet-50 and ViT-B/16 on CIFAR-100, comparing LoT to distillation methods such as BAN [23], DKD [21], and ReviewKD [22]. The teacher weights utilized in these methods were the best checkpoint for Teacher-only. The results, shown in Table 9 of the PDF, indicate that LoT outperforms these distillation baselines, underscoring the effectiveness of LoT's unique interactive learning process.
> Sure, here are the references in markdown format with each on a different row:
>
> **References:**
>
> [16] Ease-of-Teaching and Language Structure from Emergent Communication, ICLR 2019
> [17] Iterated Learning: A Framework for the Emergence of Language, Artificial Life, 2003
> [18] Iterated Learning and the Evolution of Language, Current Opinion in NeuroBiology, 2014
> [19] Cumulative Cultural Evolution in the Laboratory: An Experimental Approach to the Origins of Structure in Human Language, PNAS, 2008
> [20] Spontaneous Evolution of Linguistic Structure: An Iterated Learning Model of the Emergence of Regularity and Irregularity, IEEE Transactions on Evolutionary Computation, 2001
> [21] Decoupled Knowledge Distillation, CVPR 2022
> [22] Distilling Knowledge via Knowledge Review, CVPR 2021
> [23] Born Again Neural Networks, ICML 2018

---

> > ### Comment · Reviewer_DocJ · 2024-08-12
> >
> > I appreciate the authors' rebuttal. I raised my score from 5 to 6 in response to the authors' rebuttal. I will also discuss with the other fellow reviewers and/or AC.

---

> > > ### Author Response · Authors · 2024-08-13
> > > **Thank you for the feedback and for raising our score!**
> > >
> > > We sincerely thank the reviewer for the feedback and the score adjustment. We appreciate the reviewer's continued interest in our paper. We are committed to incorporating the new results and discussions in the revised version to further enhance the quality and contribution of our work.

---

### Official Review · Reviewer_NPF4 · 2024-07-08

**Soundness:** 2
**Presentation:** 2
**Contribution:** 3
**Rating:** 5
**Confidence:** 3

**Summary:**

This paper proposes the LOT (Learning from Teaching) regularization technique, which employs auxiliary student learners to help the main model capture these more generalizable correlations. The authors hypothesize that generalizable correlations are expected to be easier to imitate, and LOT operationalizes this concept to improve the generalization of the main model with auxiliary student learners. The results suggest the effectiveness and efficiency of LOT in identifying generalizable information.

**Strengths:**

(i) This work proposes a concept that generalizable correlations are expected to be easier to imitate.

(ii) This work proposes Learning from Teaching (LOT), a novel regularization technique for deep neural networks to enhance generalization. It is to compute a measure of ‘imitability’ for the main model to learn data correlations at the correct scales.

(iii) Multiple experiments demonstrate the effectiveness of LOT

**Weaknesses:**

(i) The core argument of this article is that "generalizable correlations are expected to be easier to imitate", but previous studies have shown that more complex data and correlations contain richer information, where models need to obtain better accuracy to learn, making them more helpful for generalization. These two views seem to be contradictory. At the same time, the author constructed their hypothesis based on human cognition, but the relevant theoretical basis is lacking, which makes its reliability questionable.

(ii) This distillation is similar to meta-learning, but its support for generalization is confused. As we all know, meta-learning performs great generalization by obtaining general knowledge by distilling task-specific knowledge and then using it to complete various tasks. But in this paper, as the author said, "both the teacher and student models may lack or possess different task-specific knowledge", it is difficult to understand why it improves generalization. I think it seems to be "joint learning", but this learning mechanism and optimization details are missing. I hope the author can further provide LOT's insight.

(iii) The description of the training process in this article is a bit hard to follow. For example, the author mentioned that generalization is improved through "joint learning", but in Section 2.2, the teacher-learner and student-learner are optimized alternately. In the introduction, the author mentioned using the student learner as an auxiliary task to improve generalization, but it is not clear how to capture the generalization connection, what this generalization connection is, and how to guide the model learning. More details may be better.

In summary, my concerns focus on the credibility of the author's motivation and the soundness of the method. If these issues can be addressed, I will be happy to improve my score.

**Questions:**

Please see the Weaknesses.

---

> ### Author Rebuttal · Authors · 2024-08-07
>
> Thank you for recognizing the novelty of our regularization method and acknowledging that our experiments demonstrate the effectiveness and efficiency of LoT in identifying generalizable information. Below, we provide detailed responses.
>
> > W1: previous studies have shown that more complex data and correlations contain richer information, where models need to obtain better accuracy to learn, making them more helpful for generalization. These two views seem to be contradictory.
>
> Thank you for pointing out the potential confusion. We appreciate the opportunity to clarify our hypothesis. Our hypothesis may be better expressed as follows: "Given a dataset D with rich enough information, if there are two learned correlations A and B that both perfectly explain D, but A is easier to imitate than B, then A is more likely to be generalizable than B."
>
> We agree that complex data containing richer information can indeed aid generalization. However, it's important to note that the complexity of the dataset and the simplicity of the learned correlations are independent factors. To clarify:
>
> 1. **Why does high complexity in the dataset help?** Real-world cases are inherently complex, so the dataset should be complex enough to capture this complexity. Otherwise, models might rely on shortcuts that don't work in real scenarios. For instance, memorizing the results of a simple addition rule, like c = a + b when a and b are within ten, is easier than understanding the rule of addition itself.
>
> 2. **Why are generalizable correlations easier to imitate given a complex dataset?** Intelligence involves finding simple, teachable rules and correlations within complex real-world data. Learning the rules of addition is simpler than memorizing 1 million addition results. Please refer to [GR2] for further explanation.
>
> In other words, AI can only emerge when an artificial learner effectively understands simple correlations, rules, and concepts of low Kolmogorov complexity that can successfully explain complex datasets. This idea, widely accepted by the deep learning community [14], can be seen as an implementation of Occam’s Razor, a principle dating back to the 14th century.
>
>
> > W1-2: the author constructed their hypothesis based on human cognition, but the relevant theoretical basis is lacking, which makes its reliability questionable. This distillation is similar to meta-learning, but its support for generalization is confused. As the author said, "both the teacher and student models may lack or possess different task-specific knowledge", it is difficult to understand why it improves generalization. I think it seems to be "joint learning", but this learning mechanism and optimization details are missing. I hope the author can further provide LOT's insight.
>
> Thank you for your insightful feedback.  For a comprehensive review of related ideas and theories in psychology and evolutionary linguistics, please refer to [GR1] and the references therein. Additionally, [GR2] provides further insights into the concept of Learning from Teaching (LoT).
>
> It is worth noting that our method is orthogonal to the meta-learning paradigm, which involves a random distribution of tasks in the meta-learning setup. The learning paradigm most closely related to our method is iterated learning, where “ease-of-teaching” is widely accepted as an important concept [15]. Our contribution can be viewed as a generalization of the iterated learning idea to broader machine learning tasks.
>
> Furthermore, please refer to [GR3] for a discussion on the challenges of developing comprehensive mathematical theories in this context.
>
>
>
> > W3: The description of the training process in this article is a bit hard to follow. The author mentioned that generalization is improved through "joint learning", but in Section 2.2, the teacher-learner and student-learner are optimized alternately. In the introduction, the author mentioned using the student learner as an auxiliary task to improve generalization, but it is not clear how to capture the generalization connection, what this generalization connection is, and how to guide the model learning. More details may be better.
>
> Thank you for your feedback. We apologize for any confusion regarding the description of the training process.
>
> To clarify, in LoT, the teacher and student are optimized alternately. The student learns from the teacher (Eq. 2), and the teacher is optimized based on both the regular task loss and the LoT regularizer, which is the feedback from the student (Eq. 1). The detailed procedure of LoT is outlined in Algorithm 1 in the paper.
>
> The generalization connection is embodied in the LoT regularizer. According to our hypothesis, a smaller imitation loss indicates a more generalizable teacher model. By incorporating the LoT regularizer as an additional loss to the teacher, the optimization process encourages the teacher to achieve a smaller imitation loss. Consequently, the generalization of the teacher model is enhanced compared to models that do not employ LoT.
>
> **References:**
>
> [14] Ilya Sutskever, An Observation on Generalization, Simons Institute, 2023
> [15] Ease-of-Teaching and Language Structure from Emergent Communication, ICLR 2019

---

> > ### Comment · Reviewer_NPF4 · 2024-08-12
> > **Official Comment by Reviewer NPF4**
> >
> > Thanks to the author for the feedback, which solved some of my confusion, but the reliability of the algorithm's insights is still not guaranteed since it basically relies on a strong Hypothesis. It would be better if a theoretical analysis could be provided. Despite this, I am still willing to revise my score from 4->5, and hope that the author can further polish and improve this work.

---

> > > ### Author Response · Authors · 2024-08-12
> > > **Thank you for the feedback and for raising our score!**
> > >
> > > We thank the reviewer for the thoughtful feedback and score adjustment! We are pleased to know that some of your confusion has been resolved.
> > >
> > > In response to your concern about the reliability of our algorithm's insights, we have provided evidence in Figure 1 of our paper to illustrate that "Generalizable correlations are more easily imitable by learners compared to spurious correlations." Additional supporting evidence is detailed in Figure 5 of the PDF in the General Response and in the discussion of Q1 in the rebuttal to Reviewer DocJ. Our hypothesis extends the concept of "ease-of-teaching" [24] from language learning to a broader range of machine learning tasks and demonstrates effectiveness in several tasks and setups.
> > >
> > > The evidence for "ease-of-teaching" in language emergence is further supported by [24], which empirically shows that "compositional language is easier to teach than a less structured language." Additionally, [25] indicates that "more compositional languages are easier to learn for new agents, including those that are architecturally different from the ones that evolved the language." We recommend referring to these works for more evidence regarding the connections between teachability (imitability) and generalization.
> > >
> > > We acknowledge that developing a comprehensive theoretical foundation for this hypothesis is indeed a challenging task, particularly in real-world contexts, and may be beyond the scope of the current paper. However, we recognize the importance of this aspect and are committed to exploring it in future work.
> > >
> > > We will include these discussions and analyses to support our hypothesis in the revised version. We genuinely appreciate your constructive suggestions and are committed to further improving the quality and contribution of our work.
> > >
> > > [24] Ease-of-Teaching and Language Structure from Emergent Communication, NeurIPS 2019
> > > [25] Compositionality and Generalization in Emergent Languages, ACL 2020

---

### Official Review · Reviewer_JrP1 · 2024-07-13

**Soundness:** 3
**Presentation:** 4
**Contribution:** 2
**Rating:** 6
**Confidence:** 5

**Summary:**

The paper introduces Learning from Teaching (LOT), a regularization technique to enhance the generalization capabilities of deep neural networks. LOT uses separate student models trained by inmate the prediction of the teacher model to provide feedback, promoting the capture of generalizable and imitable correlations. The effectiveness of LOT is demonstrated through significant performance improvements in tasks across Computer Vision, Natural Language Processing, and Reinforcement Learning, showcasing better generalization with fewer training steps compared to traditional methods.

**Strengths:**

Originality:
LOT is a novel regularization technique. This approach is unique in its hypothesis that generalizable correlations are easier to imitate, leading to the design of a regularization method that leverages the feedback from student models to improve the main model's generalization. This represents a creative combination of existing ideas from cognitive science and machine learning, applied in an innovative way to enhance neural network performance.

Quality:
The quality of the research is good. The paper have rigorous experimental validation across multiple domains, including Computer Vision, Natural Language Processing, and Reinforcement Learning. The methodology is well-detailed, with clear formulations and algorithms provided for implementing LOT in various learning contexts.

Clarity:
The paper is well-written and organized.  The introduction and motivation for LOT are clearly articulated, providing a strong rationale for the proposed approach. The methodology section is thorough, with detailed explanations and pseudocode for the LOT algorithm. The experimental results are presented with clear figures and tables, effectively illustrating the benefits of LOT.

**Weaknesses:**

1. The paper primarily compares LOT to a baseline teacher-only model. It will be the best to Include more comparisons with other regularization methods, such as dropout, batch normalization, or other recent advances in knowledge distillation, would provide a stronger validation of LOT's relative effectiveness, since it is possible that LOT could be replaced by combination of other much similar regularization.

2. Although the paper mentions the computational efficiency of LOT, a more thorough analysis of the scalability and computational cost, especially for larger models and datasets, would be beneficial. Providing detailed benchmarks on computational resources, memory usage and time required for training with LOT compared to other methods would help clarify its practical applicability in real-world scenarios.

3. The paper uses a fixed set of student models for feedback. Exploring the impact of different types of student models with varying capacities and architectures could provide deeper insights into the robustness and versatility of LOT.

4.  While the empirical results are strong, the theoretical foundations behind why generalizable correlations are easier to imitate could be elaborated further.

**Questions:**

1. I want to ask what is the performance of those NLP model or CV model on the validation set of trained datasets? Since it is also a type of generalization. It could be used to show whether the LOT is helpful for preventing over fitting.
2. For "The total training steps for LOT and the Teacher-only approach is the same for fair comparison", my understanding is that the LOT only train half number of epochs comparing to the teacher-only approach? (If we only train with one student model)
3. The paper only reports the performance the teacher model, what is performance for those students model?

**Limitations:**

I think the paper adequately addressed the limitations.

---

> ### Author Rebuttal · Authors · 2024-08-07
>
> We appreciate your recognition of the novelty and significant improvements brought by LoT, as well as the clearly articulated presentation. Below, we provide detailed responses.
>
>  > W1: It will be the best to Include more comparisons with other regularization methods, or other recent advances in knowledge distillation.
>
> As detailed in Appendix D, LoT is orthogonal to existing regularization methods such as weight decay, dropout, batch normalization, and layer normalization. For instance, in Table 1, we applied dropout to Transformer-XL models following [9]. Additionally, in Table 3, the teacher and student models are equipped with regularizations other than LoT, specifically: ViT (layer normalization, dropout, weight decay) as per the setups in [10]. LoT demonstrates effectiveness across all these settings, thereby adding value to the existing pool of regularization methods.
>
> Regarding knowledge distillation methods, we have shown that LoT outperforms the distillation method BAN [13] (Table 4). To provide stronger validation of LoT’s effectiveness, we conducted additional experiments using ResNet-50 and ViT-B/16 on CIFAR-100. We compared LoT to distillation methods such as BAN [13], DKD [11], and ReviewKD [12], with the teacher weights in these methods being the best checkpoint of Teacher-only. The results, shown in Table 9 of the PDF in GR, indicate that LoT achieves better performance than these distillation baselines, further underscoring the effectiveness of the unique interactive learning process of LoT.
>
> > W2: a more thorough analysis of the scalability and computational cost would be beneficial.
>
> We have provided a detailed comparison of the computational budget for LoT and Teacher-only in Table 12 in the PDF. Our analysis shows that LoT uses the same number of CPU cores as Teacher-only, with GPU usage being 12% to 55% higher. Despite this, LoT exhibits lower training times compared to Teacher-only (except in RL tasks) when subjected to the same total training epochs/steps, while still achieving significant performance improvements.
>
> > W3: Exploring the impact of different types of student models could provide deeper insights into the robustness and versatility of LOT.
>
> We appreciate your suggestion. In fact, our findings demonstrate that LoT remains effective across a wide range of student model architectures and capacities, as detailed in Section 3.4 and Table 3 of our paper. Key observations include:
>
> - **Enhanced Teacher Performance with Weaker Students:** For instance, a MobileNetV2 student model significantly boosts the performance of stronger models like ResNet-18 and ResNet-50 by over 1% on CIFAR-100, despite its smaller size and lower capacity.
> - **Improved Generalization with Stronger Students:** Incorporating a ResNet-50 student with a ResNet-18 teacher in LoT enhances the teacher's performance by 1.99%.
> - **Benefits of Diverse Architectures in Transformer-Based Models:** For transformer-based models such as ViTs and Swins, utilizing different architectures for the teacher and student yields better performance than using the same architecture. For example, with a ViT-B/16 student, a ViT-L/16 teacher achieves 0.27% higher accuracy compared to a ViT-L/16 student.
>
> > W4: the theoretical foundations behind why generalizable correlations are easier to imitate could be elaborated further.
>
> To provide a more comprehensive understanding, we refer you to the theoretical foundations of LoT as discussed in the fields of cognitive sciences and evolutionary linguistics in reference [GR1]. For a detailed mathematical theoretical framework, please refer to [GR3].
>
> > Q1: what is the performance of those NLP model or CV model on the validation set of trained datasets?
>
> In our experiments, we did not employ a separate validation dataset from the training dataset. We found that LoT performs effectively without the need for hyperparameter tuning on a validation set. This is particularly advantageous given the substantial resource demands associated with hyperparameter tuning on large datasets. Consequently, we believe the test performance remains reliable even without a separate validation set.
>
> To address your concern, we provided additional results on the official validation datasets for PTB and WikiText-103 in Table 10 of the PDF. These results demonstrate that LoT consistently outperforms the Teacher-only approach on both the validation and test datasets for PTB and WikiText-103, further validating the effectiveness of LoT.
>
> > Q2: my understanding is that the LOT only train half number of epochs comparing to the teacher-only approach?
>
> Yes, your understanding is correct. To maintain an equivalent total number of training steps (teacher + student) as the Teacher-only setting, LoT is trained for half the number of epochs or steps compared to the Teacher-only approach (with the exception of RL tasks). Our hypothesis, supported by results presented in Figure 1 and Figure 3 of our paper, as well as Figure 5 and Table 12 in the PDF, demonstrates that under the same total training steps, LoT converges faster and achieves better final performance compared to the Teacher-only approach.
>
> > Q3:  what is performance for those students model?
>
> **We would like to emphasize that LoT is designed primarily to enhance the generalization capabilities of the teacher model.** However, we also provided results for the student models in our experiments, as shown in Table 11 of the PDF. Our observations indicate that when the student and teacher share the same architecture, the student models can achieve performance comparable to that of the teacher models.
>
> **References:**
>
> [9] Transformer-XL: Attentive language models beyond a fixed-length context. ACL 2019
>
> [10] An image is worth 16x16 words: Transformers for image recognition at scale. ICLR 2020
>
> [11] Decoupled Knowledge Distillation. CVPR 2022
>
> [12] Distilling Knowledge via Knowledge Review. CVPR 2021
>
> [13] Born again neural networks. ICML 2018

---

> > ### Comment · Reviewer_JrP1 · 2024-08-12
> >
> > I think the authors well addressed my question. I will increase my score from 5 to 6.

---

> > > ### Author Response · Authors · 2024-08-13
> > > **Thank you for the feedback and for raising our score!**
> > >
> > > We sincerely thank the reviewer for the feedback and the score adjustment. We are pleased to hear that our response satisfactorily addressed the reviewer’s question. In the revised version of our paper, we will incorporate the new results and discussions to further enhance the quality and impact of our work.

---

### Author Rebuttal · Authors · 2024-08-07

**Highlighted General Response**

We sincerely appreciate all reviewers’ time and efforts in reviewing our paper. We also thank all reviewers for the insightful and constructive suggestions, which helped a lot in further improving our paper. In addition to our point-by-point responses below, we provide the following highlighted general responses.

**[GR1] Motivations of LoT and Related Theories in Cognitive Sciences and Linguistics:**

The concept of Learning from Teaching (LoT) originates in cognitive psychology and linguistics, particularly within the iterated learning theory of language emergence [4, 5, 6, 7]. This theory posits that the generalizable nature of languages arises from the iterative learning process across generations in a society. The core hypothesis is that a generalizable language is inherently easier to teach and learn [4, 5,6,7], which aligns with our main hypothesis.

In the AI community, recent research has aimed to employ iterated learning to enhance the generalization of emergent languages and language acquisition among artificial learners. For example, some studies have used iterated learning to improve the generalization of emergent languages between AI agents [1, 3], while others have applied it to address generalization challenges in tasks like compositional Visual Question Answering (VQA) [2].

LoT shares the same motivation as this line of research. Our primary contribution extends the concept of “ease-of-teaching” [1] from language learning to a broader range of machine learning tasks, including supervised, unsupervised learning, and reinforcement learning. We appreciate the reviewers' feedback and will incorporate these discussions into the related works and motivation sections.

**[GR2] Core Insights on Why LoT Can Improve Generalization:**

LoT functions as a regularizer, similar to other commonly used regularizers like the L2 regularizer. The L2 regularizer is effective because it encourages neural networks to learn simpler correlations, thereby avoiding overfitting. It is widely accepted that correlations with lower Kolmogorov complexity are more generalizable if they can perfectly explain a complex dataset. This aligns with the idea that "generalization equals optimal compression," as discussed by Ilya Sutskever [8]. Essentially, this notion adapts Occam’s Razor to the field of AI. Our key insight is that the "ease-of-teaching" metric serves as an effective proxy for the uncomputable Kolmogorov complexity, thus leading to a good regularizer.

Consider an intuitive example: Student A learns math by rote memorization, while Student B understands the core concepts and only memorizes essential rules, deducing the rest when needed. Both approaches can perform similarly on simple problem sets. However, as data complexity increases, Student A's burden grows significantly, while Student B's understanding-based approach remains manageable. Consequently, Student B's knowledge is easier to teach to another student, as it involves less complexity.

**[GR3] Mathematical Theory and Foundations for LoT:**

While LoT is grounded in theoretical findings from linguistics and cognitive science, we do not expect a comprehensive formal theory to emerge due to the hypothesis's dependence on specific data. In practical deep learning tasks, generalizable correlations tend to be simple enough for human comprehension. However, distinguishing between "natural" datasets and artificially complex ones is challenging.

Consider the task of learning arithmetic correlations between integers. If we have a dataset of triplets (a, b, c), there are two ground truth correlations: c = a + b and c = (a + b) mod $10^{99}$. If the dataset isn't large enough, these two correlations are indistinguishable from each other. However, our claim is that c = a + b is easier to imitate than c = (a + b) mod $10^{99}$, because the former has lower Kolmogorov complexity [8]. This claim is supported by cognitive evidence, but formalizing these intuitions mathematically—proving something holds for one correlation and not the other—is very challenging.

**References:**

[1] Ease-of-Teaching and Language Structure from Emergent Communication, ICLR 2019
[2] Iterated Learning for Emergent Systematicity in VQA, ICLR 2020
[3] Improving Compositional Generalization Using Iterated Learning and Simplicial Embeddings, NeurIPS 2023
[4] Iterated Learning: A Framework for the Emergence of Language, Artificial Life, 2003
[5] Iterated Learning and the Evolution of Language, Current Opinion in NeuroBiology, 2014
[6] Cumulative Cultural Evolution in the Laboratory: An Experimental Approach to the Origins of Structure in Human Language, PNAS, 2008
[7] Spontaneous Evolution of Linguistic Structure: An Iterated Learning Model of the Emergence of Regularity and Irregularity, IEEE Transactions on Evolutionary Computation, 2001
[8] Ilya Sutskever, An Observation on Generalization, Simons Institute, 2023

---

### Decision · Program_Chairs · 2024-09-25

**Decision:**

Accept (poster)

**Comment:**

This paper introduces LOT (Learning from Teaching), a regularization technique that uses auxiliary student learners to help the main model capture more generalizable correlations. The key hypothesis is that generalizable correlations are easier to imitate, and LOT leverages this by employing auxiliary learners to enhance the main model's generalization.

All reviewers found the proposed approach effective, supported by strong experimental results. However, the primary concern was the **validity of the hypothesis** on which this method is based. During the response period, the authors provided additional experimental evidence to validate their hypothesis, which satisfactorily addressed the reviewers' concerns.

Given the effectiveness of the approach and the comprehensive response, the paper is considered above the acceptance threshold. For the final version, it is suggested that the authors de-emphasize the hypothesis and its motivations from Cognitive Science or Linguistics. Instead, the focus should be on the demonstrated superiority of the proposed method itself, as highlighted in the ablation studies provided during the response period.